# The Impact of Falls: A Qualitative Study of the Experiences of People Receiving Haemodialysis

**DOI:** 10.3390/ijerph19073873

**Published:** 2022-03-24

**Authors:** Hannah M. L. Young, Nicki Ruddock, Mary Harrison, Samantha Goodliffe, Courtney J. Lightfoot, Juliet Mayes, Andrew C. Nixon, Sharlene A. Greenwood, Simon Conroy, Sally J. Singh, James O. Burton, Alice C. Smith, Helen Eborall

**Affiliations:** 1Leicester Diabetes Centre, University Hospitals of Leicester NHS Trust, Leicester LE5 4PW, UK; mary.harrison@uhl-tr.nhs.uk; 2Department of Respiratory Sciences, University of Leicester, Leicester LE1 7RH, UK; 3John Walls Renal Unit, Leicester General Hospital, Leicester LE5 4PW, UK; nicki.ruddock@uhl-tr.nhs.uk; 4School of Health, Wellbeing and Social Care, The Open University, Milton Keynes MK7 6AA, UK; samanthagoodliffe@yahoo.co.uk; 5Department of Health Sciences, University of Leicester, Leicester LE1 7RH, UK; courtney.lightfoot@leicester.ac.uk (C.J.L.); spc3@leicester.ac.uk (S.C.); aa50@le.ac.uk (A.C.S.); 6Department of Physiotherapy and Renal Medicine, King’s College Hospital, King’s College London, London SE5 9RS, UK; juliet.mayes@nhs.net (J.M.); sharlene.greenwood@nhs.net (S.A.G.); 7Department of Renal Medicine, Lancashire Teaching Hospitals NHS Foundation Trust, Preston PR2 9HT, UK; andrew.nixon@lthtr.nhs.uk; 8Centre for Exercise & Rehabilitation Science, Leicester Biomedical Research Unit, Leicester LE3 9QP, UK; sally.singh@uhl-tr.nhs.uk; 9Department of Respiratory Medicine, University Hospitals of Leicester, Leicester LE3 9QP, UK; 10Department of Cardiovascular Sciences, University of Leicester, Leicester LE1 7RH, UK; jb343@le.ac.uk; 11National Centre for Sport and Exercise Medicine, Loughborough University, Loughborough LE11 3TU, UK; 12Usher Institute, University of Edinburgh, Edinburgh EH8 9AG, UK; h.eborall@ed.ac.uk

**Keywords:** frailty, falls, dialysis, multi-morbidity, qualitative, interviews, experience

## Abstract

The prevalence of falls is high in people receiving haemodialysis (HD). This study aimed to explore the experiences of people receiving HD who had fallen in the last six months. A qualitative study, informed by constructivist grounded theory, used semi-structured interviews in combination with falls diaries. Twenty-five adults (mean age of 69 ± 10 years, 13 female, 13 White British) receiving HD with a history of at least one fall in the last six months (median 3, IQR 2–4) participated. Data were organised within three themes: (a) participants’ perceptions of the cause of their fall(s): poor balance, weakness, and dizziness, exacerbated by environmental causes, (b) the consequences of the fall: injuries were disproportionate to the severity of the fall leading to loss of confidence, function and disruptions to HD, (c) reporting and coping with falls: most did not receive any specific care regarding falls. Those who attended falls services reported access barriers. In response, personal coping strategies included avoidance, vigilance, and resignation. These findings indicate that a greater focus on proactively identifying falls, comprehensive assessment, and timely access to appropriate falls prevention programmes is required to improve care and outcomes.

## 1. Introduction

Chronic kidney disease (CKD) is a long-term condition characterised by abnormal kidney function and/or structure, sustained over a period of months or years [1,2]. People who have reached End-Stage Kidney Disease may require some form of renal replacement therapy (RRT) to sustain life [1]. By the end of 2019, there were approximately 68,111 people receiving RRT within the UK, of which over one-third opted to undertake haemodialysis (HD) [3]. HD is a maintenance treatment that is typically prescribed thrice weekly for four hours throughout the patients’ lifespan or until successful transplantation [4]. HD is equivalent to approximately 10% of kidney function and cannot replace all their functions [4]. Subsequently, people receiving HD also require several additional interventions to meet defined treatment targets for calcium, phosphate, and blood pressure [4] and to manage anaemia [5] and ongoing dietary interventions to counter the impact of chronic fluid overload, hyperphosphataemia, and malnutrition [6].

Despite the intensity of treatment, HD is associated with a range of adverse health outcomes. Falls, defined as ‘unexpectedly coming to rest on the ground, floor, or lower-level’, are common in the haemodialysis (HD) population [7]. The incidence of falls ranges between 1.2 and 1.6 falls/patient-year, compared to 0.6 to 0.8 falls/patient-year in the general older population [8,9]. Falls are a predictor of mortality in people receiving HD and are also associated with reduced mobility, independence, quality of life, and poorer transplant outcomes [10,11,12,13,14,15]. Falls can also lead to injury, including increased fracture risk due to renal osteodystrophy, as well as hospital admissions, adding to an already high level of healthcare utilisation in this population [12,16,17,18].

Falls always have multifactorial contributory factors, but of particular importance in the HD population is an increased prevalence of sarcopenia and frailty [19,20], and balance deficits [9,21,22]. People receiving HD often live with complications from multiple long-term conditions and the effects of polypharmacy, which are associated with increased falls risk [23,24,25]. Further important factors are behavioural and psychological (such as the feelings of mastery and confidence or fear of falling) and environmental (for example, uneven flooring and ill-fitting footwear) [13,26]. Postural sway and falls risk scores are also significantly increased following HD treatment, and it is hypothesised that this is the consequence of fluid shifts and electrolyte imbalances, intradialytic hypotension, prolonged immobilisation, and fatigue [8,9,12,18,27,28,29]. Reduced cerebral blood flow during HD has also been linked with impaired cognition in CKD which may contribute to falls [30,31].

The management of falls would benefit by being tailored to the requirements of the individual, accommodating their needs and preferences, and addressing potential barriers to accessing support [32]. Consequently, a detailed understanding of the experiences and impact of falls upon people receiving HD is crucial. To date, research has focused on observational research, therefore the purpose of this paper is to explore experiences of people receiving HD who had recently fallen. Specifically, we aimed to understand the circumstances precipitating the fall(s), the immediate and long-term impacts of the fall, and the strategies used to cope with falling.

## 2. Materials and Methods

This paper presents findings from a study that focused on the experiences of people receiving HD who were also living with frailty. The analysis we present concentrates on participants’ experiences of falls. Ethical approval was granted by the NHS Research Ethics Committee South West (Bristol; REC ref: 17/SW/0048). We used the consolidated criteria for reporting qualitative research (COREQ) to report the study [33].

### 2.1. Participant Selection

We recruited adults who had been receiving HD for at least three months from three haemodialysis centres within the East Midlands Renal Network. Participants were recruited from three haemodialysis centres within the East Midlands in the UK between May 2017 and January 2019. Eligible participants were able to provide informed consent, spoke English, had a history of at least one fall in the last six months, and were classified as vulnerable to severely frail according to the Clinical Frailty Scale (CFS score 4–7) [34]. We initially used maximum variation sampling to recruit a diverse range of participants, following this with theoretical sampling to recruit those with characteristics that could extend the preliminary themes developed [35]. All participants were provided with an information sheet and the study was explained in full.

### 2.2. Data Collection

We used semi-structured interviews and solicited diaries to gather data. HMLY, HE, and a patient and public involvement (PPI) group convened for the study and developed a topic guide for the interviews which was based on literature but also informed by experience. We piloted and refined this guide during the first three interviews. The final version is included within Appendix A.

Interviews were conducted by HMLY, who was known to some of the participants, although not directly involved in their clinical care. Interviews took place during HD or in the participant’s home, dependent on their preference. We used field notes to record initial reflections after each interview.

Following their interviews, we asked each participant to complete a diary gathering contemporaneous information about their experiences of falls over a period of up to three months. We offered them the option to record a written or audio diary, which allowed those with visual impairments, difficulties writing, or a limited command of written English to participate [36]. We provided them with a ‘diary-keeping pack’. This included an information booklet developed with the PPI group (outlined in Appendix A), the researcher’s contact information, and, depending on participant preference, an A5 notepad and pen, or an audio recorder and instructions. Participants had full control over the content of their diary, and the frequency and depth with which they recorded their experiences [37]. We regularly reminded them to complete their diary throughout the diary-keeping phase, via the phone or face-to-face contact on the HD unit, depending on participant preference [37,38]. Data collection ceased at the point at which new data no longer generated additional insight.

### 2.3. Data Analysis

Interviews were audio-recorded and then transcribed verbatim and anonymised. We used NVivo Software (QSR International, Version 11, London, UK) to manage the analysis. Analysis was informed by the principles of constructivist grounded theory, which involved several stages [35]. Firstly, the transcripts of the initial ten interviews and four diaries were read and re-read by HMLY. Line-by-line coding was used to develop initial themes, which were then discussed with the wider research team. HMLY, MH, and NR then familiarised themselves with the remaining transcripts and coded these according to the themes developed. We used a constant comparison to compare new data with that previously collected to adapt, expand, or merge codes and/or themes. Finally, we identified those themes most relevant to the research aims [35]. Throughout all stages, we used memos to engage with the data and create an analytical audit trail [35].

### 2.4. Methods to Ensure Quality and Rigour

High-quality qualitative research is delineated by rigor, sincerity, and credibility [39]. Rigour in our approach to sampling is demonstrated by our extensive prior experience of working with this population, and two years within the field gathering the data. We used maximum variation sampling initially to ensure a diverse range of participants were included and followed this with theoretical sampling to recruit participants who were able to provide new data that was used to check, expand and extend the themes constructed. We carried out data collection until we collectively felt that data saturation in relation to the research question, where new data no longer generates new insight or extends themes identified, had been achieved. The identification of this point was facilitated by the process of constant comparison, where insights from new data prompted the review of interviews and diaries collected previously, and further theoretical sampling [35].

Sincerity was achieved via reflexivity which helped the lead researcher (HY) to recognise how her background, experience, and knowledge may have influenced the qualitative research process and the interpretation of the findings [35,39]. She used a reflexive journal to document these influences and to create an ‘audit trail’ of analytical decisions and the development of codes and categories. Throughout the research, we discussed our personal views on participants’ experiences. We also presented our findings with the wider multidisciplinary research team to enhance credibility.

The credibility of the analysis and validity of the findings was enhanced by using multiple data sources (interviews and diaries), which provided a form of methodological triangulation [39]. In addition, two researchers (NR/MH) independently coded a subset of the transcripts. Whilst the codes were not expected to be identical, reflecting on the level of agreement and similarity between coders served as a form of researcher triangulation, enabling us to ensure that our coding was grounded in the data [39]. Thick description of the data and detailed quotations demonstrate the credibility and consistency of the study’s findings. Finally, whilst we did not use member checking, the results of this study were presented to PPI members and their feedback was used to enhance credibility [39].

## 3. Results

Of 37 individuals approached, 26 provided written informed consent; one died prior to data collection, leading to a final sample of 25. Interviews lasted 30 to 120 min (mean 62 min). Participant characteristics are outlined in Table 1. They reported a median of three falls (IQR 2–4) within the preceding six months. Eleven participants were living with a spouse or partner, nine alone, and five with extended family. Interviews predominantly took place at the HD unit (23, 92%). Thirteen participants agreed to complete a diary, and twelve declined. Ten diaries (five written and five audio-recorded) were returned for analysis. Diaries were kept for a mean of 86 ± 24 days and a mean of 4 ± 7 entries were recorded. Of those participants who kept diaries, five recorded no falls, three a single fall, and two reported two falls over the diary-keeping period. Of the seven falls reported via the diaries, three resulted in injury (one a laceration, one bruising, and one musculoskeletal pain).

To present the findings, we begin by describing the context in which the falls occurred. Participants’ experiences were then organised with three themes: (a) perceptions of the cause of their fall(s), (b) the consequences of the fall, and (c) decisions about reporting the falls and the strategies used to cope with increased falls risk. The themes are presented alongside illustrative quotations within Table 2, Table 3 and Table 4.

### 3.1. Context of the Fall, and Perceptions of the Cause

Most participants had experienced multiple falls, and most often fell in a forward-backward direction. While some participants fell in the period immediately following HD or during a hospital admission, most falls occurred at other times, predominantly within the home.

The main intrinsic factors they believed had led to their falls were poor balance, weakness, and dizziness, exacerbated by environmental hazards. Loss of balance when transferring from sitting to standing, walking, reaching up, bending down, and negotiating the stairs was common. For some, this was worsened by neuropathy which reduced their co-ordination, ability to feel and sense where their feet were, and to clear the ground whilst walking. Undertaking two tasks simultaneously was also associated with falls, and participants noted that it was important to maintain their concentration to keep their balance. They attributed the increasing weakness to the initiation of HD and increasing time on HD when they were required to be inactive for treatment. Whilst dizziness was often experienced following haemodialysis, for some, it was more generally related to postural changes and medications. Others reported symptoms such as vertigo, instability on turning, and altered spatial awareness. These intrinsic impairments were exacerbated by extrinsic environmental hazards.

### 3.2. Consequences of the Fall

Participants described feelings of shock and pain immediately following their falls. Distress and fear were also common, related to injuries and the potential to be admitted to hospital. Those who fell outside, and one participant who fell when her front door was open and was unable to summon help, were worried for their immediate safety. Embarrassment was also common, and men were particularly concerned that people might assume that they were intoxicated.

Most participants described long periods of time lying on the floor after being unable to get up independently. They primarily relied upon family members or neighbours help, but those who lived alone recounted how they persevered until they were able to get up, knowing they would have to rely on the ambulance service if they could not.

Injury was common, particularly cuts and bruises. One participant described dislocating her shoulder, another tearing the muscles around her shoulder, and others exacerbating existing musculoskeletal problems or damaging joint replacements following a fall. Several participants reported fractures affecting both the upper and lower limbs, ribs, and vertebrae. The extent of the injury was often disproportionate to the severity of the fall described, with a ‘small stumble’ leading to multiple fractures. Healing was often protracted or incomplete. Some participants explained how their falls had also affected their haemodialysis treatment, including having to have a tunnelled central venous catheter inserted for HD treatment when their fistula arm was plastered following a fracture. One participant had avoided this by having a ‘window’ made within her cast. Others described bleeding and bruising from their fistula following a fall, requiring investigation and intervention from their renal team.

The longer-term ramifications of injury were pain, reduced range of movement, and loss of function. Participants described requiring family assistance or support from paid carers, reducing independence. Others became unable to drive and were more isolated as a result. Participants also highlighted how falling had negatively impacted their confidence. They were highly fearful of future falls with worse consequences and questioned their ability to undertake tasks or participate in important social events.

### 3.3. Reporting and Coping with Falls

Most participants had reported their falls to a healthcare professional, typically a member of their renal team or their GP. Participants perceived this disclosure as being met with disinterest and reported not being provided with any specific advice or follow-up care. Participants who chose not to report their falls explained basing this decision upon an assumption that falls were an expected consequence of HD treatment and advancing age, their fault, and that nothing could, or would, be done. Others did not report their falls out of concern that they would be admitted to the hospital. Only one person reported being proactively asked about falls. Those who had been referred to services for fallers reported mixed experiences. Those who had attended a falls prevention programme had found it beneficial, but some had been told they were not appropriate, and others could not attend because of scheduling clashes with HD.

Participants were generally resigned to the fact that they would continue to experience falls. This attitude was closely linked to a view that falls were a normal consequence of ageing and HD, reinforced by the responses from healthcare professionals when they reported their falls. Those who fell as inpatients described becoming ‘out of practice’ with walking and personal care through limited opportunity. This was felt to be compounded by staff not encouraging activity due to a perceived lack of time and concerns regarding safety. One participant noted how these strategies might increase the likelihood of falling in the future. In response, their primary coping strategies were avoidance of tasks that they associated with an increased risk of falling and vigilance during tasks that could not be circumvented. They carefully planned tasks they perceived to be riskier, monitoring for signs and symptoms which had previously preceded a fall and increased their reliance upon aids and adaptions to avoid future falls. In addition, participants also spoke of slowing down and reducing their activity as a protective mechanism. Several participants who had experienced a fracture described inventive approaches to coping with reduced upper limb function, including switching to favour the non-dominant side, using their body to stabilise items to manipulate with the non-dominant side, and using everyday objects creatively to get a challenging task done more easily.

## 4. Discussion

This study aimed to explore the experiences of people receiving HD who had recently fallen. In parallel with the general older population, they experienced multiple falls with multifactorial causes. The immediate consequences of the falls were shock, pain, long periods of lying down or being unable to get up, and feelings of fear and embarrassment, and falls were inconsistently reported to healthcare professionals. In contrast to the general older population, people receiving HD sustained injuries that were disproportionate to the fall experienced, resulting in fractures, long-term musculoskeletal impairments, and disruption to HD treatment. Very few participants reported being proactively asked about falls and those who did found that little action was taken, or experienced barriers to attending falls services. In response, resignation, avoidance, and vigilance were adopted as their primary coping strategies, leading to loss of confidence and independence. The results of this study suggest that the management of falls in the HD population may be improved via proactive identification and a comprehensive assessment, in addition to overcoming barriers to accessing interventions and services.

Ensuring the people receiving HD are regularly asked about falls as a routine part of their kidney care is a fundamental first step to improving care in this population. Falls were inconsistently reported to healthcare professionals and very few participants were proactively asked about falls. Incorporating the prospective recording of falls within the routine practice in the dialysis unit is feasible, would take the onus off the patient, and enable clinicians to consistently identify those who require further assessment and input [28,40]. The findings of this study indicate that, once a fall is reported, a comprehensive falls history should be obtained, prevention advice delivered, and onward referral for more comprehensive assessment and treatment made as indicated. This necessitates appropriate training amongst nephrologists and nurses and routine access to occupational therapists and physiotherapists as a part of the kidney care team. Despite this, access to therapist support remains low within the UK [41,42].

Our findings also suggest that not all people receiving HD who fall will be persuaded to discuss their falls. The reasons given for not reporting falls echo those of older people within the general population, who also believed falls to be beyond their personal control, unavoidable, or were fearful that acknowledging a fall would threaten their independence [43,44]. *How* falls are explored and discussed may be important amongst those reluctant to report their falls. Some participants were alarmed by the information they received from health professionals, which exacerbated their avoidance and hypervigilance. Exploring individual values and beliefs about fall risk, perceived barriers, and facilitators to fall prevention strategies may encourage greater disclosure [45]. This would allow healthcare professionals to challenge any preconceptions regarding the inevitability of falls, negotiate acceptable falls prevention strategies and address any identified barriers to participation in falls prevention programmes [45]. Furthermore, emphasising falls prevention interventions to maintain independence and build social connections may be a more effective method of encouraging participation than emphasising the prevention of falls or falls-related injury [44,45,46].

Those who did report their falls found that very little action was taken, or experienced barriers to attending falls services. As a result of these experiences, they were resigned to further falls and used avoidance and vigilance as their primary coping strategies. Many of these coping strategies reflect those adopted by older people within the general population, who implement them to gain a sense of control [43,44,46]. Such strategies have, however, been linked to further weakness, mobility limitations increasing the risk of future falls and, as was the case within our study, social isolation, and loneliness [44]. These issues may be particularly problematic in the HD population who already have a high burden of sarcopenia, frailty and treatment, and low levels of social participation [47,48,49]. Our findings indicate that more effective strategies for falls prevention in the HD population are urgently required.

Falls were predominantly attributed to weakness and poor balance, which may be amenable to exercise. To date, only a limited number of small studies have explored the impact of exercise on falls and falls-related outcomes in the HD population. The majority of these have focused upon intradialytic exercise, which comprises a range of heterogeneous forms of exercise delivered during HD treatment. Intradialytic cycling, delivered by means of a cycle ergometer may reduce the incidence of falls, but does not appear to influence fear of falling [40,50]. Intradialytic resistance training and intradialytic cycling, in combination with exercises delivered outside of the dialysis session or nutritional supplementation, appear to improve balance and reduce falls risk [21,51,52,53]. Only one of these interventions has included progressive balance training [53]. The results of our study underline the importance of being able to respond appropriately to environmental hazards, which were frequently identified as contributors to falls, suggesting that a greater emphasis should be placed on high-level balance training in exercise interventions designed for this population. Indeed, strength and balance training are known to significantly reduce the rate and number of falls in community-dwelling older people [54].

Participants’ accounts of vertigo, instability on turning, hearing changes, and altered spatial awareness preceding a fall also highlight the potential contribution of vestibular dysfunction. Vestibular dysfunction often goes unrecognised in older people and may be more pronounced within the HD population, where symptoms could more readily be attributed to hypotension or autonomic dysfunction [25,55]. Limited evidence indicates that the incidence of vestibular dysfunction increases with advancing CKD severity [56]. Those receiving HD are particularly susceptible due to exposure to otoxic medications, uraemia, neuropathy, anaemia, electrolyte imbalance, hypotension, and the impact of metabolic derangements, all of which can negatively affect the vestibular system [56,57,58,59]. Future research examining the contribution of vestibular dysfunction to falls appears to be warranted in this population.

Many of the rehabilitation needs described are addressed in falls prevention programmes widely available to older people within the general population. One barrier facing people with HD appears to be a lack of access to these programmes due to scheduling clashes and a perception that people receiving HD are not appropriate amongst healthcare professionals. Scheduling clashes may be overcome by increased availability and flexible programmes, but also offering home-based, online or telephone supervised programmes. These alternative forms of delivery are acceptable to older people in the general population and have been associated with enhanced participation, but care is required to ensure safety and to deliver the social aspects of face-to-face group programmes that are an important motivator to participation [44,45]. Our study also indicates that work needs to be done to educate and build links with healthcare professionals delivering falls prevention programmes to address any preconceptions about the suitability of people receiving HD.

To our knowledge, this study is the first to explore the experiences of people receiving HD who have had a recent fall. Key strengths were the use of sampling approaches which enabled an ethnically diverse range of individuals to participate. Despite this, falls diaries were only kept by half of the participants which limited our exploration, although this was offset by the simultaneous use of interviews [38]. Some participants described experiences that indicated underlying cognitive impairment, for example, difficulties with dual tasks, and our understanding of this would have been strengthened by a measure of cognitive function.

## 5. Conclusions

In summary, falls in the HD population are multifactorial leading to wide-ranging negative consequences. Poor reporting and barriers to accessing appropriate care led people receiving HD to adopt potentially unhelpful coping strategies. Care may be improved by proactively questioning about falls, conducting a comprehensive assessment, and carefully discussing falls risk. Whilst tailored strategies for falls interventions may be warranted in the HD population, improving access to the effective falls prevention programmes readily available to older people in the general population would be a useful first step to improving falls-related outcomes in this population.

## Figures and Tables

**Table 1 ijerph-19-03873-t001:** Participant Demographics. Data are mean ± standard deviation or median (IQR) unless otherwise indicated.

		*N* = 25
Age (years)		69 ± 10
Gender *n* (%)	Female	13 (52%)
	Male	12 (48%)
Ethnicity *n* (%)	White background	13 (52%)
	Asian or Asian British	10 (40%)
	Caribbean	1 (4%)
	Not stated	1 (4%)
Cause of CKD	Diabetic nephropathy	11 (44%)
	Aetiology uncertain	6 (24%)
	Chronic pyelonephritis	3 (12%)
	Atypical hemolytic uremic syndrome	1 (4%)
	Focal segmental glomerulosclerosis with nephrotic syndrome	1 (4%)
	Henoch-Schonlein Purpura	1 (4%)
	Minimal change nephropathy	1 (4%)
	Polycystic kidney disease	1 (4%)
Charlson Co-morbidity Index		6 ± 2
Time on haemodialysis (months)		43 (16–85)
Clinical Frailty Scale (CFS) score *n* (%)	CFS 4, Vulnerable	9 (36%)
	CFS 5, Mildly frail	5 (20%)
	CFS 6, Moderately frail	8 (32%)
	CFS 7, Severely frail	3 (12%)
Number of falls in last six months		3 (2–4)
Previous transplant *n* (%)	No	21 (84%)
	Yes	4 (16%)
Active on transplant list *n* (%)	No	22 (88%)
	Yes	3 (12%)
Employment status *n* (%)	Retired	21 (84%)
	Unemployed	3 (12%)
	Part-time employed	1 (4%)
Marital Status *n* (%)	Married	15 (60%)
	Single	5 (20%)
	Widowed	5 (20%)
Social Circumstances *n* (%)	Lives with spouse or partner	11 (44%)
	Lives alone	9 (36%)
	Lives with extended family	5 (20%) ^1^

^1^ Demographic characteristics were extracted from participants’ medical records. Ethnicity categories were taken from NHS ethnicity coding within the medical notes. Clinical Frailty scoring was undertaken by the participants’ consultant nephrologist and information on falls and social circumstances were gathered from the participants.

**Table 2 ijerph-19-03873-t002:** Quotations illustrating participants’ perceptions of the cause of their falls.

**Intrinsic Factors Believed to Lead to Falls**	**Weakness** *“I fell down because I didn’t have enough power in my legs. You can fall automatically; you don’t know if you have got no strength.”* (Participant 6, male, Asian British, age 50s). *“My legs went shaky, then my knees buckled, and I fell over.”* (Participant 14, female, White British, age 80s).
	**Dizziness** *“It’s like being on the ferry with the roll of the waves…It’s almost like your centre of balance has moved…As you lurch you’re just kind of “wow!” Things just spin a bit, and you have to be careful.”* (Participant 16, male, White British, age 60s). *“I was at the sink, and I reached up to get my tablets out and I looked up and I looked down and then all of a sudden I started to fall back. But I managed to grab the sink. I would like to be able to stand at the sink and not…feel as though I am going backwards although I am not moving.”* (Participant 4, female, White British, age 60s).
	**Balance** *“I have not managed to fall for quite a while, its only if my mind is distracted and I am not concentrating then I am prone to a fall.”* (Participant 22, female, White British, age 60s). *“I do know how to walk, it’s just the balance. If you told me to close my eyes and take two steps I would probably not be able to do it because I know as soon as I close my eyes I am going to fall, I can’t keep my balance at all…”* (Participant 5, male, White British, age 60s).
**Extrinsic factors believed to lead to falls**	**Environmental hazards** *“I skidded and then I fell, I fell so many times there because of the snow.*” (Participant 7, female, Asian British, age 80s). *“I fell on the bus…I was at the back … trying to sit down and [the driver] jarred like that and I went down the bus and landed on the steps backwards, oh it was horrible”’* (Participant 9, female, White British, age 60s).

**Table 3 ijerph-19-03873-t003:** Quotations illustrating the immediate and long-term consequences of falling.

**Immediate Consequences of the Fall**	**Shock** *“I was sitting outside in the wet, cold, freezing weather on the floor. I couldn’t stand up because for about half an hour, you go into shock because of the extreme pain. So, you are sitting on the floor until you get the confidence to stand up.”* (Participant 5, male, Asian British, age 60s).
	**Pain** *“When you have a fall it is always painful. But what do you do? you just have to put up with it. When I had a fall for the shoulder they gave me ibuprofen gel, it didn’t work. So, it took them about 3 or 4 days before one of the doctors says we will give you 2.5 milligram of Oramorph, I said what is 2.5 going to do? He says that’s the highest, my hands are tied because of your kidneys.*” (Participant 5, male, Asian British, age 60s).
	**Fear** *“I was so wobbly that when I went to unlock my front door I fell and knocked the back of my head. I was flat on my back for hours. I fell with the front door open I never got to lock it. So, I was like oh my god I can’t fall asleep. I could not black-out because basically if I had I could have woken up to a house empty”* (Participant 3, female, Black British, age 40s). *“When you fall down it really gets you. You know nobody’s there to help you. It is scary. You have to take extra care, I’ve got no one to look after me. So, it does scare me.”* (Participant 6, male, Asian British, age 50s).
	**Embarrassment** *“When I fell on the escalator I felt stupid with everybody crowding round me.”* (Participant 26, male, White British, age 80s). *“If you find that you cannot get up and you have got to hang on to something to pull yourself up that’s when people see you struggling. But if they just see you lying there they acquaint that with a bloody drunk. People assume so much that’s completely wrong.”* (Participant 25, male, White Irish, age 70s).
	**Being unable to get up and long lies** *“I knew I was falling but I thought I’ll be able to push myself up again. But I didn’t. I tried to get up and I couldn’t. After an hour I wriggled backwards to some boxes in the kitchen doorway and tried to push myself up from those and I couldn’t. I wriggled backwards into the lounge and tried to get up at the lounge entrance onto things but couldn’t do it. I was so weak and dizzy, and I was trembling after all the effort of trying to get up. I phoned an ambulance, and they came along. I’d managed to get up and was sitting on the edge of the bed then, but this is three and a half hours from when I first fell.”* (Participant 16, male, White British, age 60s).
	**Fractures and other injuries** *“The first fall I could err walk a bit, the second one I had to learn how to walk again. I fractured my err right leg and ruptured all the muscles in my shoulder.”* (Participant 15, female, White British, age 80s).
	**Effect on haemodialysis treatment** *“The first time I had fallen I had broken this hand so they operated and put pins in and a plaster on, but they cut a window with the fistula, so it could be used. The second time I fell and broke my arm my needle site was covered so then they did the vascath.”* (Participant 10, female, Asian British, age 50s).
**Long term consequences of the fall**	**Loss of function and mobility** *“I haven’t been out anywhere you know. My son does the shopping and all that and I feel isolated, I want to go, I want to go to church, I want to go to my, my heart says to go but my legs will not take me.”* (Participant 7, female, Asian British, age 80s).
	**Lack of confidence** *“If something happens you are always wary of that happening again. So, I have been aware, and it has affected my confidence. In your mind [you think] I going to fall, and you are more or less suggesting to yourself that you will fall.”* (Participant 20, male, White British, age 70s).

**Table 4 ijerph-19-03873-t004:** Quotations illustrating participants’ attitudes to the reporting of falls and how they cope with falling.

**Reporting of Falls**	**Not reported** *“I just put it down to my age and my middle ear problem you know. I would not even think [to tell an HCP], they’d probably say “no we don’t know about that blah blah blah.”* (Participant 4, female, White British, age 60s). *“I do not think I did tell anyone about my fall to be quite honest. I never thought they would be interested. That is not decrying what they do, I just thought it was my fault and so what can they do about it.”* (Participant 12, male, White British, age 70s).
	**Reported, no action** *“When I told [the nurse] she said, ‘Oh there is nothing much we can do, go and tell your GP.’ That is it.”* (Participant 2, male, Asian British, age 70s). *“When I told them I had a fall they did not want to know. They said, ‘you are perfect, your [blood] levels are perfect.*” (Participant 6, male, Asian British, age 50s).
	**Reported and referred to falls services** *“I have wondered whether an exercise programme would help but I have recently been to the falls clinic, and they told me that there is nothing they could do for me, that I am too poorly to be included in an exercise class. I find that quite disappointing actually.”* (Participant 13, male, White British, age 50s). *“I would go to a [falls prevention] programme but there is appointments and three days I am on dialysis.”* (Participant 6, male, Asian British, age 50s).
**Coping strategies**	**Resignation** *“Nothing much can be done about this [the falls] …. there is nothing that can be changed, it has been done to the best.”* (Participant 1, male, Asian British, age 60s). *“As there is nothing I can do to help, I must take a philosophical view and get on with life.”* (Participant 8, female, White British, age 60s). **Avoidance** *“The doctor says if you fall on the concrete you are going to break your hip and it is going to shatter…What can you do? You can sit in a wheelchair and stop [yourself] from falling.”* (Participant 5, male, Asian British, vulnerable, age 60s). *“A few years back I went to get out the car because I felt weak and horrible and collapsed in the gutter. I am not fussy where I land. But now I do not go out for that reason.”* (Participant 22, female, White British, age 60s).
	**Vigilance** *“I want to walk fast but I can not, I have to think of my safety.”* (Participant 17, female, Asian British, age 60s). *“In the house I know there are certain areas that I can kind of throw myself on to and then turn over and sit up. So, I know I can throw myself on to a chair, I know I can throw myself on to the settee. I can throw myself on to the bed because I know I can get up from there.”* (Participant 13, male, White British, age 50s).
	**Slowing down** *“I have always been a bit of a quick walker so… when these things have happened, I am like OK time to slow down.”* (Participant 3, female, Black British, age 40s). *“I just move so slow, so I have reduced that risk.*” (Participant 22, female, White British, age 60s). *“Since the last fall I have slowed down, the crutches are there and I am just being a bit more careful and cautious.”* (Participant 16, male, White British, age 60s).
	**Adaptation** *“We went to [the shop] Tuesday and he [husband] said…“I am getting you this, it is a cake spatula.” I said, “I do not do flipping cakes, do you think I am going to start doing cakes now, one-handed?” He said, “No, it is to help you spread the bread in the morning”.”* (Participant 4, female, White British, age 60s. This participant had fractured her arm following a fall and describes a creative method for getting around the resultant functional limitations that she experienced). *“Even now after I broke it and everything, I never really carried anything with his hand anyway. Look it does make a difference that it is broken to carry, to even… but I have adapted.”* (Participant 10, female, Asian British, age 50s. This participant had fractured her arm following a fall).

## Data Availability

The data presented in this study are available on request from the corresponding author.

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
