# Peer review of "The Impact of Falls: A Qualitative Study of the Experiences of People Receiving Haemodialysis"

_ijerph, 2022, doi:10.3390/ijerph19073873_

Round 1

Reviewer 1 Report

Presented study is descriptive one. However the Authors it as "observational research". Paper looks like a report (not a research paper!)

?? a study which focused on the experiences of people receiving HD who …..

What does mean “constructivist grounded theory”? The Authors should deeply revise and update the paper before the next submission

Findings should be presented in a more specific way.

Objective measurement are missing

Criteria of classifications are missing

What does mean acronymous HMLY? HE?

“2.3. Data analysis” should be described in a more specific way.

What does mean “themes”?

The major concern is in which way the Author cut off all subjective interpretations? The purpose of the research paper is an objective knowledge!

Author Response

Presented study is descriptive one. However the Authors it as "observational research". Paper looks like a report (not a research paper!)

The paper reports a qualitative research study, which was conducted using robust qualitative research methodology by experienced researchers. Elements of the findings are indeed descriptive and, moreover, analytical. The study reported was not ‘observational research’. Indeed, towards the end of the introduction we point out that: “To date, research has focused on observational research, therefore the purpose of this paper is to explore experiences of people receiving HD who had recently fallen.”

This is to differentiate our paper, a qualitative exploration of patients’ experiences, from previous research, which has all been observational. We do this to identify the important gap in the literature, which this paper addresses.

 What does mean “constructivist grounded theory”? The Authors should deeply revise and update the paper before the next submission

Constructivist grounded theory is a well-recognised approach to qualitative methodology, one of the variations from the founders’ (Glaser and Strauss) initial grounded theory in the 1960s. A grounded theory approach aims to move the analysis beyond a description of experiences to a more conceptual level. There have been many developments within the field of grounded theory, and differing standpoints offered by its proponents. The constructivist standpoint acknowledges that theory does not develop organically but is shaped by context and the influence of the researcher. Rather than viewing the researcher as a distant and objective observer, they are part of the process, being explicit and reflective about their unique experience and perspectives and how this may have shaped data collection and analysis. For more information we direct the reviewer to the work of Kathy Charmaz.

In the manuscript we provide a detailed account of the stages of analysis as follows:

Firstly, the transcripts of the initial ten interviews and four diaries were read and re-read by HMLY. Line by line coding was used to develop initial themes, which were then discussed with the wider research team. HMLY, MH, and NR then familiarised themselves with the remaining transcripts and coded these according to the themes developed. We used constant comparison to compare new data with that previously collected to adapt, expand, or merge codes and/or themes. Finally, we identified those themes most relevant to the research aims[35]. Throughout all stages we used memos to engage with the data and create an analytical audit trail[35].”

And the sampling methods unique to grounded theory as follows:

“We initially used maximum variation sampling to recruit a diverse range of participants, following this with theoretical sampling to recruit those with characteristics which could extend the preliminary themes developed [35]

We believe this is ample detail for readers to determine if the approach used was rigorous and appropriate. As constructivist grounded theory is a well-established qualitative approach, we do not feel it is necessary to provide a lengthy description of ‘what’ this is.

Findings should be presented in a more specific way.

Findings are presented in a comprehensive manner consistent with reporting guidance for qualitative research (see Tong A, Sainsbury P, Craig J. Consolidated criteria for reporting qualitative research (COREQ): a 32-item checklist for interviews and focus groups. International journal for quality in health care. 2007 Dec 1;19(6):349-57.)

 Objective measurement are missing

This research uses a qualitative approach, and accordingly there a no ‘objective measurements’. We do however, provide ‘objective’ information on the sample characteristics in Table 1.

 Criteria of classifications are missing

Please can you specify what is meant by ‘criteria of classifications’.

 What does mean acronymous HMLY? HE?

HMLY refers to Hannah Mary Louise Young (the first author) and HE to Helen Eborall (the senior author). These acronyms signify who conducted specific elements of the work and are a requirement of COREQ reporting (see above reference).

“2.3. Data analysis” should be described in a more specific way.

The description of analysis provided outlines all the analytical steps taken and complies with qualitative reporting guidance as previously referenced.

What does mean “themes”?

 ‘Themes’ are features of participants’ accounts characterising perceptions and/or experiences that the researcher sees as relevant to the research question. The development of themes are a typical feature of qualitative research, and it is unlikely this will need explaining to an audience familiar with qualitative research methodology.

The major concern is in which way the Author cut off all subjective interpretations? The purpose of the research paper is an objective knowledge!

As detailed above, this is a qualitative research paper which adheres to the COREQ reporting guidelines. Qualitative research does not seek to achieve ‘an objective knowledge’. Rather, a strength of qualitative research is to explore, analyse and foreground people’s experiences. Qualitative research does not seek to make any formal generalisations as would be the case with quantitative research. Instead, studies such as ours aim to achieve resonance, that is, to help the reader make connections between the themes or findings in the study at hand, and intuitively apply them to their own contexts.

Reviewer 2 Report

Hannah et al reported a qualitative study of the experiences of falls of people receiving haemodialysis. The authors presented the data of twenty-five adult participants who had recently fallen within three themes (a) perceptions of the cause of their fall(s), (b) the consequences of the fall and (c) decisions about reporting and coping with falls. The study is interesting and would be of interest to those working in the field. 

I have some concerns for the study:

1) The participants in the study are mainly older adults, whereby falls are common due to aging. There are a number of different risk factors associated with falls. The authors need to assess the risk factors comprehensively to determine whether the falls are really the consequence of receiving haemodialysis. For example, it is important to check all patients if they have fallen in the past before the HD treatment as a predictor of their risk. The study would benefit from more systematic evaluation of the underlying cause of the falls rather than based merely on the participants' perceptions.

2) The time of fall is not reported. Does it often occur immediately after HD? Is there any pattern of the fall occurrence?

3) In the results, the sentence "They lasted 30 to 120 minutes (mean 62 minutes)", what does it refer to?

Author Response

The participants in the study are mainly older adults, whereby falls are common due to aging. There are a number of different risk factors associated with falls. The authors need to assess the risk factors comprehensively to determine whether the falls are really the consequence of receiving haemodialysis. For example, it is important to check all patients if they have fallen in the past before the HD treatment as a predictor of their risk. The study would benefit from more systematic evaluation of the underlying cause of the falls rather than based merely on the participants' perceptions.

Thank you for your comment. This would be of critical importance if this study was an observational study. It is not our aim to determine whether the falls are really the consequence of receiving haemodialysis. Our aim is to explore experiences of people receiving HD who had recently fallen. As such understanding which risk factors contribute to falls in this group is not important to this study and has in fact been addressed in previous quantitative research. We acknowledge that falls are multifactorial, and are interested in participants overall experiences of falling, the consequences of these for the person and their experiences of reporting them, irrespective of what caused them.

The time of fall is not reported. Does it often occur immediately after HD? Is there any pattern of the fall occurrence?

Again, because of the aims of this research, we are interested in participants’ experiences of falling. Because of this focus and because of the sample size (which is appropriate for qualitative research) we do not, nor cannot, make any inference about patterns of falls. This would necessitate a quantitative research design.

In the results, the sentence "They lasted 30 to 120 minutes (mean 62 minutes)", what does it refer to?

We refer to the length of the interviews and have changed the text to make this clearer.

Reviewer 3 Report

It is an article with a certain originality, mainly in terms of the information provided by hemodialysis patients on the measures taken after the falls suffered, both by these patients and by the healthcare professionals who were informed, showing a need on the part of the medical services of greater proactive measures, in addition to a greater follow-up of those patients who have suffered falls.
On the other hand, the method used is correct, as well as the structure of the article. As for the results, they are easy to interpret and are displayed appropriately, with consistent conclusions based on the data obtained.

Regarding the possible improvements of general concepts, the following are detailed:
- Improve the specification of the objective of the study (page 2), it only mentions "to explore experiences of people receiving hemodialysis who had recently failed", but I think that it should be more specific to what type of experiences the research is focused on.
- In point 2 “Materials and methods” (page 2) the method and material used in the study should be described, as well as I think that the following information is not adequate at this point: “This paper presents findings from a study which focused on the experiences of people receiving hemodialysis who were also living with frailty. The analysis we present concentrates on participants' experiences of falls”.
- The date and period in which the data collection was carried out must be included.
- A section detailing the procedure followed to endow the study with trustworthiness must be included in the method. I enclose the reference of two recently published qualitative investigations that include a reliability section and that can serve as an example:
* Idowu, O.; Makhinova, T.; Quintanilha, M.; Yuksel, N.; Schindel, T. J.; Tsuyuki, R.T. Experience of Patients with COPD of Pharmacists’ Provided Care: A Qualitative Study. Pharmacy 2021, 9, 119.
* Muthuri, R.N.D.K.; Senkubuge, F.; Hongoro, C. Senior Managers’ Experience with Health, Happiness, and Motivation in Hospitals and the Perceived Impact on Health Systems: The Case of Meru County, Kenya. Healthcare 2021, 9, 350.
- Finally, and although it is not a relevant point, including the detailed COREQ Checklist as supplementary material would be very valid additional information (optional improvement).

Regarding the possible improvements of specific aspects, an error is observed in the date shown in the ethics statement on page 11 (Bristol; REC ref: 17/SW/0048 20th of February 2022), because the date of study approval is a date after the submission of this article. To conclude, the bibliographic reference number 35 is distributed in 3 different lines and can be grouped in a single line.

To conclude, congratulations to the authors of this article for the chosen research topic and I encourage them to continue contributing to improving the quality of life in hemodialysis patients.

Author Response

- Improve the specification of the objective of the study (page 2), it only mentions "to explore experiences of people receiving hemodialysis who had recently failed", but I think that it should be more specific to what type of experiences the research is focused on.

Thank you. We have amended as follows:

‘ …the purpose of this paper is to explore experiences of people receiving HD who had recently fallen.’ Specifically, we aimed to understand the circumstances precipitating the fall(s), the immediate and long-term impacts of the fall, and the strategies used to cope with falling.’

- In point 2 “Materials and methods” (page 2) the method and material used in the study should be described, as well as I think that the following information is not adequate at this point: “This paper presents findings from a study which focused on the experiences of people receiving hemodialysis who were also living with frailty. The analysis we present concentrates on participants' experiences of falls”.

Thank you for your suggestion. We describe the materials and methods used at length in subsections 2.1-2.3, which is all included under the ‘materials and methods’ heading.

The text to which you refer is some introductory information, which highlights that the results of this paper arise from a broader exploration of the experiences of people living with frailty and HD. This work is currently under consideration in another publication; hence we are not able to reference it here at this time.  Because there was too much information specifically relating to falls to include in detail in that paper, we focus on falls in this specific analysis.

This introductory sentence is not intended to be all the material and methods – that information is outlined in detail in sections 2.1-2.3 as previously stated.

- The date and period in which the data collection was carried out must be included.

Thank you for your comment. We have added the following information into the manuscript Participants were recruited between May 2017 and January 2019.”

- A section detailing the procedure followed to endow the study with trustworthiness must be included in the method. I enclose the reference of two recently published qualitative investigations that include a reliability section and that can serve as an example:

Thank you for sharing these examples. We have added the following section at the end of the manuscript as requested:

“Methods to ensure quality and rigour

High-quality qualitative research is delineated by rigor, sincerity, and credibility (39).  Rigour in our approach to sampling is demonstrated by our extensive prior experience of working with this population, and two years within the field gathering the data. We used maximum variation sampling initially, to ensure a diverse range of participants were included, and followed this with theoretical sampling to recruit participants who were able provide new data that was used to check, expand and extend the themes constructed. We carried out data collection until we collectively felt that data saturation in relation to the research question, where new data no longer generates new insight or extends themes identified, had been achieved. The identification of this point was facilitated by the process of constant comparison, where insights from new data prompted the review of interviews and diaries collected previously, and further theoretical sampling (35).  

Sincerity was achieved via reflexivity which helped the lead researcher (HY) to recognise how her background, experience and knowledge may have influenced the qualitative research process and the interpretation of the findings (35,39). She used a reflexive journal to document these influences and to create an ‘audit trail’ of analytical decisions and development of codes and categories. Throughout the research, we discussed our personal views on participants’ experiences. We also presented our findings with the wider multidisciplinary research team to enhance credibility.

Credibility of the analysis and validity of the findings was also enhanced by using multiple data sources (interviews and diaries), which provided a form of methodological triangulation (39). In addition, two researchers (NR/MH) independently coded of a subset of the transcripts. Whilst the codes were not expected to be identical, reflecting on the level of agreement and similarity between coders served as a form of researcher triangulation, enabling us to ensure that our coding was grounded in the data (39). Thick description of the data, and detailed quotations, demonstrates the credibility and consistency of the study's findings. Finally, whilst we did not use member checking, the results of this study were presented to PPI members and their feedback used to enhance credibility [39].”

To the acknowledgements we also add the following information, to further clarify:

“The researchers responsible for analysis of the data consisted of four female researchers (HY,NR,MH,HE). HY is a physiotherapist working with people with long-term conditions. At the time of the interviews, she was undertaking a mixed-methods PhD and had previous experience of leading qualitative research. NR is a renal dietitian, and MH a diabetes specialist nurse, and both have experience in conducting qualitative research. HE is a social scientist, with over 20 years of qualitative research experience.”

Finally, and although it is not a relevant point, including the detailed COREQ Checklist as supplementary material would be very valid additional information (optional improvement).

Thank you, we submit the COREQ checklist as supplementary material suggested.

Regarding the possible improvements of specific aspects, an error is observed in the date shown in the ethics statement on page 11 (Bristol; REC ref: 17/SW/0048 20th of February 2022), because the date of study approval is a date after the submission of this article.

Thank you for noticing this error, we have now updated to the following: specific aspects, an error is observed in the date shown in the ethics statement on page 11 (Bristol; REC ref: 17/SW/0048 20th of February 2017).

To conclude, the bibliographic reference number 35 is distributed in 3 different lines and can be grouped in a single line.

Thank you for noticing this. We spotted this at the initial submission phase and were assured by the journal that this would be corrected if the article was accepted. However, we have tried to correct in this updated version of the manuscript.